# Effect of Irrigation, Nitrogen Fertilization and Amino Acid Biostimulant on Proximate Composition and Energy Value of *Pisum sativum* L. Seeds

Wioletta Biel [1,*], Cezary Podsiadło [2], Robert Witkowicz [3], Jagoda Kępińska-Pacelik [1] and Sławomir Stankowski [4]

[1] Department of Monogastric Animal Sciences, Division of Animal Nutrition and Food, West Pomeranian University of Technology in Szczecin, Klemensa Janickiego 29, 71-270 Szczecin, Poland
[2] Department of Agroengineering, West Pomeranian University of Technology in Szczecin, Juliusza Słowackiego 17, 71-434 Szczecin, Poland
[3] Department of Agroecology and Crop Production, University of Agriculture in Krakow, Mickiewicza 21, 31-120 Krakow, Poland
[4] Department of Agroengineering, West Pomeranian University of Technology in Szczecin, Papieża Pawła VI 3, 71-459 Szczecin, Poland
* Correspondence: wioletta.biel@zut.edu.pl

**Abstract:** The study investigated the impact of biological and agricultural conditions on the chemical composition and energy value of pea seeds for poultry feed. In the experiment, the species assessed was pea (*Pisum sativum* L.), a determinate form, cultivar (cv.) Cysterski. During the field experiment, the response of peas to the following experimental factors was assessed: first factor—irrigation (yes, no), second factor—nitrogen fertilization (0, 20 and 40 kg N·ha$^{-1}$), third factor—amino acid biostimulant (yes, no). In both years of experiments, representative seed samples were collected, in which the chemical composition was assessed for the content of dry matter, crude protein, crude fat, crude fiber, and nitrogen free extract. Additionally, in order to assess the suitability of the evaluated pea seeds as poultry feed, the energy value expressed in the form of apparent metabolizable energy (AME$_n$) was estimated. The protein content in pea seeds increased after the application of the biostimulant, both with and without irrigation. Statistically significant differences in the crude fiber content in pea seeds were found under the influence of the interaction of irrigation and nitrogen fertilization. The use of the biostimulant significantly increased the energy value of pea seeds.

**Keywords:** biostimulant; estimated apparent metabolizable energy; irrigation; macronutrients; nitrogen fertilization; pea seeds

## 1. Introduction

In modern poultry fattening, the basis of nutrition is complete feed mixtures, the nutritional value of which must be adapted to the growth rate and protein deposition potential. High demand for protein means that, apart from cereals, high-protein feeds are the basic feed component of mixtures. The deepening negative balance of high-protein feed materials is covered by soybean meal (SBM). It should also be noted that the demand for protein in farm animal nutrition has increased. Domestic production incl. legumes, including peas, lupines (white, yellow, blue), field beans, and soybeans, cover about 30% of the protein requirement. The remaining 70% comes from imported SBM [1]. About 83% of global annual soybean production consists of genetically modified seeds [2]. In Europe, as well as globally, there has been an ongoing debate for many years about whether and to what extent genetically modified feeds, especially soybeans, can be eliminated from animal nutrition [3,4]. Legume seeds are an interesting alternative to soybeans [5]. Legume plants play an important role in modern agriculture, which is due in part to their ability to symbiosis with papillary bacteria and the resulting use of nitrogen from the

air [6]. Species in this family are grown for food and fodder in all regions of the world, and their nutritional value and beneficial effects on the yield of successor crops are widely documented. Legume seeds are valuable for their high nutritional value, but the possibility of using them in organic animal production is limited due to their processing such as fat extraction with organic solvents. Unfortunately, the importance of these crops has declined in Europe, while soybean acreage has increased on other continents [7]. The demand for feed protein in Europe is mainly covered by genetically-modified soybean meal (SBM GM). Due to the growing reluctance of consumers to genetically modified organisms [8] in order to reduce the content of SBM GM in feed, in recent years there has been an intensive search for alternative sources of vegetable protein with a similar content of nutrients. It seems that legume seeds have the greatest potential [5]. The available literature shows that legume seeds contain a relatively large amount of protein [9,10]. For environmental and nutritional reasons, increasing the cultivation area of legumes becomes important [11]. The production of legume seeds is one of the priorities to increase protein security in the European Union [12,13]. Another factor driving research into alternative sources of protein is the constantly growing price of soybean meal [14]. Pea (*Pisum sativum* L.) is one of the most important legumes in the world, with global annual production estimated at roughly 13.5 million tons and a producer price of over \$200/ton, and at present it is grown in over 90 countries [15]. Currently, in Europe, among the legume species, mainly pea is cultivated [16]. Pea seeds and their products are a major source of protein for humans as well as animals [5,17]. The amino acid composition of pea seeds is similar to SBM [18,19]. The nutritional value of the protein of all species of legume seeds, including peas, is obviously lower than the protein of SBM. When evaluating the nutritional value of the protein of these two feed sources (per 100 g of protein), it can be seen that the pea seeds' amino acids composition is similar to SBM [20]. Pea seeds contain amounts of lysine close to soybean meal (7.29 and 6.23 g·100 g$^{-1}$ protein, respectively) and methionine (1 and 1.43 g·(100 g)$^{-1}$ protein, respectively), which are necessary for poultry or pig diets [21,22]. Standardized Ileal Digestibility (SID) values are an important indicator that can help nutritionists formulate broiler diets that better meet the birds' requirements and minimize excess nutrients. For lysine, the SID is similar for both pea and SBM (92% and 91%, respectively) [23,24]. Livestock can be fed peas without significant loss of carcass quality [25]. Pea seeds can be a valuable source of protein in poultry mixes, especially of broiler, laying-hens, chickens and turkeys diets [26]. For the desired results, the recommended proportion of these seeds in complete feed mixtures should not be exceeded. The proportion of pea seeds in complete mixtures for poultry depends on many factors, including the concentration of antinutrients, protein quality, energy concentration, age of the birds, or the direction of production (egg production, reproduction, fattening) [27–29]. Egg quality is determined based on many traits important for global egg production, and depends on many factors, including the diet and age of hens. Koivunen et al. [30] showed that it is possible to use peas in the diet of laying hens with no negative impact on the quality of eggs. The chemical composition of the seeds of different pea varieties is different and is differentiated by the cultivation technology, i.e., the level of fertilization, the cultivation and plant protection system, and the cultivation environment [31]. There are real opportunities for significant independence from imported soybean products, but it is important to keep in mind that pea yields are variable, influenced by agrotechnical and habitat factors, especially weather. One of the basic conditions for obtaining high yields of good quality is to provide plants with optimal conditions for growth and development during the growing season. Pea yields are highly dependent on rainfall deficits and extreme air temperatures [32]. Ensuring high fertility requires proper fertilization [33,34]. Also, nitrogen fertilization can affect yields, as well as the level and nutritional value of pea seed protein. Erman et al. [35] observed the beneficial effects of mineral nitrogen on yielding and growth, as well as protein content and yield of pea seeds. In addition to allowing plants to grow and develop properly, irrigation also stimulates the supply of nutrients to plants [36]. Pea is the genus with high water requirements during growth, pod setting and seed filling [37]. Irrigation lengthens the growing

season, but consequently results in higher yields [38]. Current technological advances aim to increase yields and improve their quality while minimizing environmental risks. Among the industrial inputs in modern agriculture are plant-growth biostimulants. The use of biostimulants is an effective way to reduce the adverse effects of environmental stresses on plants. A plant biostimulant is any substance or microorganism applied to plants with the aim to enhance nutrition efficiency, abiotic stress tolerance and crop quality traits, regardless of its nutrient content [39]. Regulation of the European Parliament and Council (EC) defines plant biostimulants as "EU fertilizing product able to stimulate plant nutrition processes independently of the product's nutrient content with the sole aim of improving one or more of the following characteristics of the plant or the plant rhizosphere: (1) nutrient use efficiency, (2) tolerance to abiotic stress, (3) quality traits, or (4) availability of confined nutrients in the soil or rhizosphere" [40]. These products are environmentally safe and contribute to sustainable, high-output low-input crop productions [41,42]. The most commonly used biostimulants are based on humic compounds, amino acids or seaweed extracts [43]. Amino acid biostimulants are a way to increase plant resistance and the effectiveness of traditional soil fertilization. Mixtures of amino acids and peptides are obtained by chemical and enzymatic hydrolysis of proteins from agro-industrial by-products, both from plant sources (crop residues) and animal wastes (e.g., collagen, epithelial tissues) [44,45]. Chemical synthesis can also be used for single or mixed compounds. Other nitrogenous molecules include betaines, polyamines and non-protein amino acids. Providing plants with additional amino acids reduces the energy input required to assimilate nitrogen. Research highlights the usefulness of biostimulants in the cultivation of many crops worldwide [44,46,47]. When added to soil, they increase fertility by increasing nitrogen (N) uptake and assimilation, chelating heavy metals, and acting as hormone-like molecules, or due to their antioxidant activity depending on the specific amino acid containing products [48]. Researchers primarily emphasize their effects on plant yield, but also on the quality of the raw material [49–51]. However, there is still little information on the effect of amino acid biostimulants on the yield and nutritional value of plants, especially legume crops. Increasing the pea cultivation area and improving the yield stability of this species could contribute to a significant reduction in the shortage of protein feeds. The research hypothesis assumed an increase in pea seed yield and improved nutritional and energy quality under the influence of spraying with biostimulant with amino acids, irrigation, and nitrogen fertilization on the yield and proximate composition of pea seed cv. Cysterski.

## 2. Materials and Methods

### 2.1. Field Experiment

In the years 2020–2021, a field experiment was carried out in the split-split-plot design in three repetitions. The area of the experimental plot was 40 m$^2$ and the area of the harvest plot was 32 m$^2$. The field experiments were carried out in Lipnik (53°41′ N, 14°97′ E) at the Agricultural Experimental Station belonging to the West Pomeranian University of Technology in Szczecin (Figure 1). Poland is located in the temperate climate zone. The soil belongs to light loamy sand, with weakly loamy sand underneath and, in some places, light silt. Typologically, it is characterized as brown soil according to the World Reference Base for soil resources [51].

Pea seed, a self-terminating form of the cv. Cysterski, was treated against fungal pathogens (Maxim 025 FS, Syngenta Poland) and inoculated with *Rhizobium* bacteria (Nitragina, BIOFOOD, Wałcz, Poland).

The first experience factor (two levels) was the use or not to use irrigation. Irrigation was applied based on the rainfall needs of peas. When the actual rainfall was lower than the rainfall needs by at least 10 mm, irrigation was carried out with a dose of 20 mm due to low soil retention (light soil).

The second factor was nitrogen fertilization applied before sowing on three levels: 0, 20 and 40 kg N·ha$^{-1}$. Fertilization was applied in the form of ammonium nitrate.

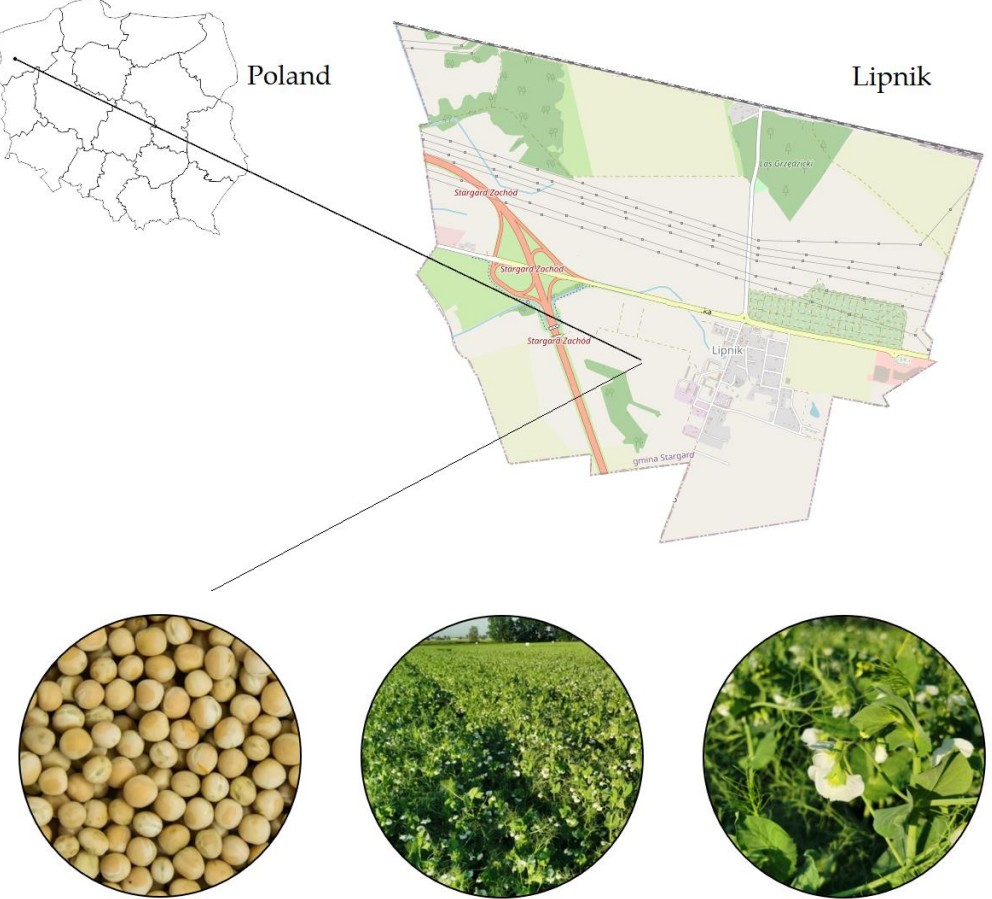

**Figure 1.** The map of the collection region.

The third factor (two levels) was the use or not of an amino acid biostimulant. The biostimulant Aminoplant (Siapton® ISAGRO, Italy) was used in the experiment. The biostimulant contained: total nitrogen (N)—9.1%, organic nitrogen ($N_{org}$)—8.7%, ammonium nitrogen (N-NH$_4$)—0.4%, free amino acids (FAA$_S$)—10.0%, and organic carbon ($C_{org}$)—24%. The biostimulant contained 18 L-amino acids derived from the hydrolysis of animal proteins. The biostimulant was applied twice during the growing season at a dose of 1.5 L·ha$^{-1}$. The first treatment was performed at the beginning of inflorescence development (BBCH 51), and the second one at the end of phase BBCH 55 in the form of a fine spray (300 L of water per ha).

Tillage and maintenance treatments were carried out in accordance with generally accepted agrotechnical recommendations. The yield of field pea seeds, harvested from the field in the full maturity phase, was given at 14% moisture content.

### 2.2. Chemical Analyses

Samples of seeds were ground to 0.1 mm with a laboratory mill type KNIFETEC 1095 (Foss Tecator, Höganäs, Sweden). The chemical composition of samples was determined according to the Association of Official Analytical Chemists [52] procedures: dry matter, by drying at 105 °C to a constant weight; crude fat (as ether extract), by Soxhlet extraction with diethyl ether; crude ash, by incineration in a muffle furnace at 580 °C for 8 h; crude protein (N × 6.25), by Kjeldahl method, using a Büchi Distillation Unit B−324 (Büchi Labortechnik AG, St. Gallen, Switzerland), crude fiber was determined with a fiber analyzer—ANCOM$^{220}$ (ANCOM Technology, Macedon, NY, USA); nitrogen-free extract (NFE) was calculated as follows: NFE (%) = 100 − % (moisture + crude protein + crude fat + crude ash + crude fibre).

### 2.3. Energy Value

The energy value of pea seeds expressed in the form of AME$_n$ (nitrogen-corrected apparent metabolizable energy) for poultry was calculated based on Equation (1) [53]:

$$\text{AME}_n \text{ (MJ·(1000 g)}^{-1}) = 0.1803 \times \text{CP}_{\text{digestible}} + 0.0388 \times \text{EE}_{\text{digestible}} + 0.0173 \times \text{NFE}_{\text{digestible}} \quad (1)$$

where: CP, crude protein; EE, ether extract (crude fat); NFE, nitrogen-free extract

The digestibility coefficients for pea seed digestibility calculations given for poultry feedstuffs in the European Table of Energy Values (ETEV) were used to calculate the digestible component, amounting to 86% for CP, 80% for EE and 77% for NFE [53].

### 2.4. Statistical Analyses

Three factorial analysis of variance (ANOVA) was carried out using the STATISTICA v. 13.3 software (TIBCO Software Inc., Palo Alto, CA, USA) [54]. Below is a mathematical model of the analysis of the variance of the experiment assumed in the split-split-plot design (Table S1).

$$y_{ijlp} = m + a_i + g_j + e_{ij} + b_l + ab_{il} + e_{ijl} + c_p + ac_{ip} + bc_{lp} + abc_{ilp} + e_{ijlp}$$

where:

$y_{ijlp}$—value of the examined feature for the *i*-th level of factor A, *l*-th level of factor B, *p*-th of factor C in the *j*-th repetition,
$m$—mean of the experimental setup,
$a_i$, $b_l$, $c_p$—the effects of the studied factors, respectively,
$g_j$—replication effect,
$ab_{il}$, $ac_{ip}$, $bc_{lp}$, $abc_{ilp}$—appropriate effects of factor interaction,
$e_{ij}$, $e_{ijl}$, $e_{ijlp}$—random effects.

The Tukey's Honestly Significant Difference (HSD) Test at *p* = 0.05 was used to find the differences between means.

## 3. Results and Discussion

Keeping poultry in optimal condition is one of the most important tasks of breeders. The diet and living conditions of the animals are the two main factors affecting the condition of the poultry. Adequate nutrition covers the birds' needs for nutrients that fulfill their role in the growth process, maintenance of homeostasis, and metabolism [55]. Properly balanced (in terms of composition and nutrient content) feed is one of the basic elements determining breeding and production success. Current breeding progress in peas is based on a more complete understanding of the plant's biology, and is expected to result in the possibility of improving cultivation technologies. In the presented own research, the use of a biostimulant (Table 1) had a significant impact on pea seed yield. Its introduction into the technology increased seed yield by about 13%.

Amino acids are well-known biostimulants that have a positive effect on plant growth and yield [44]. They have the advantage of mobility and easy transport in plants [56]. Amino acids can directly or indirectly affect plant growth and yield. The use of irrigation did not cause a significant increase in yield, but a tendency to increase it can be stated (*p* = 0.096). Singh et al. [57] obtained higher grain yield after applying irrigation and nitrogen fertilization only at a dose of 15 kg·ha$^{-1}$. Research on pea irrigation management is limited. The experience carried out so far shows that the yield-forming role of rainwater irrigation is no less than that of mineral fertilization, especially since the highest yield increases as a result of irrigation were obtained under conditions limiting the effectiveness of fertilization, i.e., in dry years on light soils [57]. Paredes et al. [58] showed that water saving is possible without significant impact on pea yield. There is a need to better understand the factors influencing water use and water-yield relations that may be helpful to improve irrigation management with an emphasis on quality for food industry. Nitrogen

fertilization at 20 kg·ha$^{-1}$ also showed a tendency to increase grain yield ($p$ = 0.08) (Table 1), as confirmed by previous studies [59,60]. Nitrogen has many functions in plant life. Being responsible for the biosynthesis of amino acids, proteins, nucleic acids, chlorophyll, and various primary and secondary metabolites [61]. One of the reasons for low pea yields may be mineral nitrogen deficiencies, with concomitant disruption of coexistence with *Rhizobium*, e.g., due to poor papillae as a result of foraging for abalone. Positive effects of nitrogen on yield were reported by Singh et al. [62] and Yadav et al. [63]. Voisin et al. [64] found that variable yields of pea seeds are caused by the content of N$_{min}$ in the soil and mineral nitrogen fertilization affecting the activity of root nodules. Positive effects of amino acid biostimulant application were found on many vegetables [49,65,66], but there is no information about the effect of this biostimulant on pea seeds. In experiments with sugar beet, potato, strawberry, cereals, tomatoes, or squash, positive effects of the amino acid biostimulant on the yield of these crops were found [65–67]. On the other hand, the research by Kunicki et al. [68] did not clearly confirm the beneficial effect of the biostimulant on the yield of plants. Different crop species are characterized by different susceptibilities to the stress factor and the associated reduced yield.

**Table 1.** Yield (Mg·ha$^{-1}$), proximate composition (%) and metabolizable energy (MJ·(1000 g)$^{-1}$) of the evaluated pea seeds.

| Factor | Level | Mean * | *p* (HSD) |
| --- | --- | --- | --- |
| Grain yield | | | |
| Irrigation | Yes | 4.252 [a] | 0.096 (1.132) |
| | No | 3.464 [a] | |
| Nitrogen fertilization | 0 kg·ha$^{-1}$ | 3.669 [a] | 0.080 (0.637) |
| | 20 kg·ha$^{-1}$ | 4.199 [a] | |
| | 40 kg·ha$^{-1}$ | 3.706 [a] | |
| Biostimulant | No | 3.608 [a] | 0.017 (0.392) |
| | Yes | 4.108 [b] | |
| Dry matter | | | |
| Irrigation | Yes | 92.035 [a] | 0.053 (0.079) |
| | No | 91.959 [a] | |
| Nitrogen fertilization | 0 kg·ha$^{-1}$ | 91.986 [a] | 0.612 (0.081) |
| | 20 kg·ha$^{-1}$ | 92.013 [a] | |
| | 40 kg·ha$^{-1}$ | 91.992 [a] | |
| Biostimulant | No | 92.003 [a] | 0.518 (0.040) |
| | Yes | 91.991 [a] | |
| Crude ash | | | |
| Irrigation | Yes | 2.933 [a] | 0.068 (0.167) |
| | No | 2.792 [a] | |
| Nitrogen fertilization | 0 kg·ha$^{-1}$ | 2.885 [a] | 0.124 (0.050) |
| | 20 kg·ha$^{-1}$ | 2.858 [a] | |
| | 40 kg·ha$^{-1}$ | 2.845 [a] | |
| Biostimulant | No | 2.888 [b] | <0.001 (0.018) |
| | Yes | 2.837 [a] | |
| Crude protein | | | |
| Irrigation | Yes | 23.729 [a] | <0.001 (0.070) |
| | No | 25.283 [b] | |
| Nitrogen fertilization | 0 kg·ha$^{-1}$ | 23.103 [a] | <0.001 (0.092) |
| | 20 kg·ha$^{-1}$ | 25.136 [b] | |
| | 40 kg·ha$^{-1}$ | 25.279 [c] | |

**Table 1.** *Cont.*

| Factor | Level | Mean * | *p* (HSD) |
|---|---|---|---|
| Biostimulant | No<br>Yes | 23.856 [a]<br>25.155 [b] | <0.001 (0.055) |
| Crude fat | | | |
| Irrigation | Yes<br>No | 1.198 [a]<br>1.188 [a] | 0.193 (0.021) |
| Nitrogen fertilization | 0 kg·ha$^{-1}$<br>20 kg·ha$^{-1}$<br>40 kg·ha$^{-1}$ | 1.181 [a]<br>1.196 [a]<br>1.202 [a] | 0.229 (0.033) |
| Biostimulant | No<br>Yes | 1.123 [a]<br>1.263 [b] | <0.001 (0.018) |
| Crude fibre | | | |
| Irrigation | Yes<br>No | 5.017 [a]<br>5.488 [b] | 0.012 (0.226) |
| Nitrogen fertilization | 0 kg·ha$^{-1}$<br>20 kg·ha$^{-1}$<br>40 kg·ha$^{-1}$ | 5.545 [c]<br>5.201 [b]<br>5.013 [a] | <0.001 (0.095) |
| Biostimulant | No<br>Yes | 5.347 [b]<br>5.159 [a] | <0.001 (0.064) |
| NFE | | | |
| Irrigation | Yes<br>No | 59.158 [b]<br>57.291 [a] | <0.001 (0.153) |
| Nitrogen fertilization | 0 kg·ha$^{-1}$<br>20 kg·ha$^{-1}$<br>40 kg·ha$^{-1}$ | 59.272 [c]<br>57.622 [a]<br>57.779 [b] | <0.001 (0.117) |
| Biostimulant | No<br>Yes | 58.789 [b]<br>57.660 [a] | <0.001 (0.122) |
| AME$_n$ | | | |
| Irrigation | Yes<br>No | 11.95 [b]<br>11.93 [a] | 0.006 (0.491) |
| Nitrogen fertilization | 0 kg·ha$^{-1}$<br>20 kg·ha$^{-1}$<br>40 kg·ha$^{-1}$ | 11.86 [a]<br>11.96 [b]<br>11.98 [c] | <0.001 (0.431) |
| Biostimulant | No<br>Yes | 11.90 [a]<br>11.98 [b] | <0.001 (0.354) |

*—means within a given source of variation marked with the same letters do not differ statistically; NFE, nitrogen−free extract; AME$_n$, nitrogen-corrected apparent metabolizable energy; HSD, honestly significant difference.

There was no effect of irrigation, nitrogen fertilization, and the applied biostimulant on the dry matter content of pea seeds, although there is a tendency to reduce its content after irrigation. The current direction of pea cultivation aims, among other things, at increasing the protein content in the seeds. Hacisalihoglu et al. [69] showed a broad range from 12.6% to 33.1% for peas seeds depending on the genetic factor and growing conditions. This allows promoting the cultivation of this crop and ensures that it can be used in the feed industry, including poultry feeding. In our work, the protein content of seeds increased significantly after applying nitrogen fertilization compared to the protein content of seeds from the control, which confirms that the protein content of peas is a function of the N status of the soil [70]. Hu et al. [71] showed that low nitrogen application increases nitrogen concentration in peas. The application of irrigation significantly reduced the protein content of the seeds tested, which can be explained by dilution effects.

The application of the biostimulant caused a significant increase in protein concentration in pea seeds compared to the control. The range of protein concentration in pea seeds was nearly 2% (23.103–25.279%), which is consistent with data presented by other authors [72,73]. It is worth noting, however, that the minimal level of protein indicated by the authors was lower (minimum mean 15%). This reveals the need for better varieties and agronomic practices that can increase the protein concentration in pea seeds. Sterna et al. [74] showed that farming systems (organic, conventional) have no significant effect on seed protein content. Their overall results (of a five-year analysis) showed that the protein content of pea ranged from 20.0% to 26.1%. As shown by Uhlarik et al. [75] cultivating for increased seed protein content is hampered by the negative correlation between protein content and yield. Therefore, increasing protein production can be conducted by increasing the area under protein crops or by improving protein quality over protein content, given that differences in protein content can be the result of many different environmental factors. Peas have a relatively low lipid content compared to other legumes [76]. Of the factors analyzed, only the application of the biostimulant statistically significantly increased the crude fat level in pea seeds compared to that found in seeds from the control facility by 12.5% (Table 1).

Dietary fiber is a feed component of plant origin that is digested to a small extent by monogastric animals, but a certain amount is necessary for the proper functioning of the digestive tract. This component affects the nutritional value of feed and animal health [77,78]. Crude fiber is a cell wall fraction composed of cellulose and hemicelluloses encrusted with lignin. Nutritional standards still only consider the content of crude fiber as one of the basic criteria for the nutritional value of compound feed components for individual poultry species taking into account age and use. For the bird's body, the correct supply of this component is very important because fiber is not just a ballast substance in the feed. In addition to typical carbohydrate feeds such as cereals (corn, wheat, barley), which in poultry nutrition form the basis of their diet [79] and are a source of crude fiber, peas can also provide these components. Compared to other legume seeds, peas have the least fiber in their composition [76,80,81]. All the treatments evaluated had a significant effect on the crude fiber content of pea seeds ($0.001 < p < 0.012$). The application of nitrogen fertilization resulted in a decrease in crude fiber content compared to that found in seeds from the control facility. The application of irrigation caused a significant decrease in fiber levels from 5.488% to 5.017%. The average crude fiber content found in the tested pea variety was 5.25%, which confirms the research of other authors [82].

In the composition of legume seeds, nitrogen-free extract (NFE), unlike cereals, do not make up the main proportion of pea seed dry matter. NFE accounts for an average of 58%. A significant effect of the factors studied on the content of these components was found.

Currently, the main components of the diet for broilers are corn (about 65% of metabolizable energy) and SBM as the main source of protein. However, in order to improve the efficiency of the poultry ration, it is necessary to know the exact energy value of the individual feed components, since the energy balance of the diet can determine the intake of nutrients and thus affect the performance and characteristics of the broiler carcass. Pea seeds can also serve as an energy feedstock in poultry feed due to the highest metabolizable energy content among all seeds of legumes [83–85]. The estimated value of $AME_n$ based on equations was 11.94 MJ·$(1000\ g)^{-1}$ of seeds. The factors studied had a significant effect on the level of $AME_n$. It is noteworthy that the energy value of pea seeds increased significantly after the application of the biostimulant (Table 1).

Figures 2–5 show the influence of the interaction of research factors on the content of nutrients in pea seeds. From the data presented, it can be seen that protein content increases after application of the biostimulant, both with and without irrigation, but a greater increase in the content of this component was determined on the irrigated object by more than 1.6% (Figure 2). The interaction of the factors of biostimulant and nitrogen fertilization was also statistically significant (Figure 3).

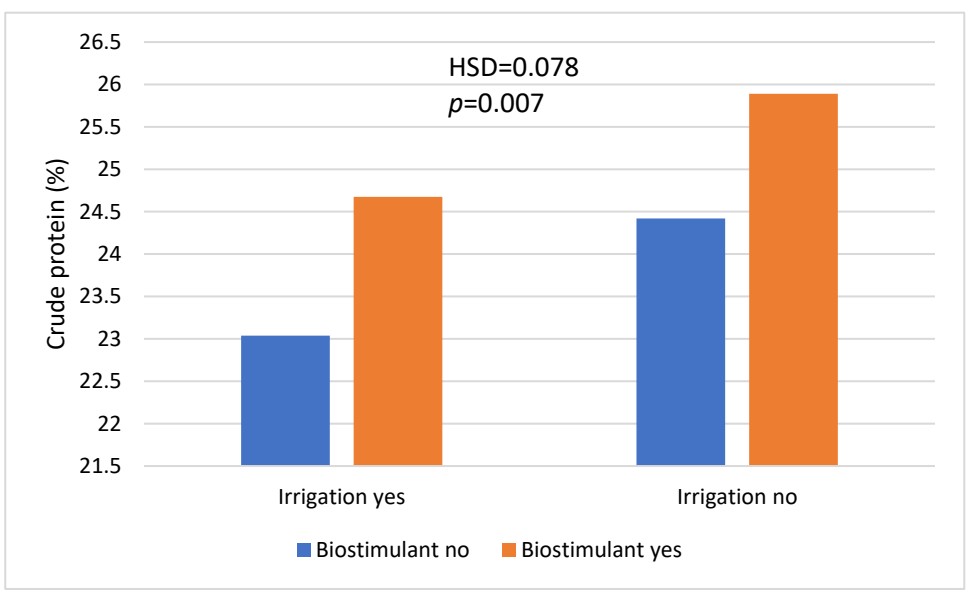

**Figure 2.** Crude protein content in pea seeds after application of irrigation and biostimulant. HSD, honestly significant differences.

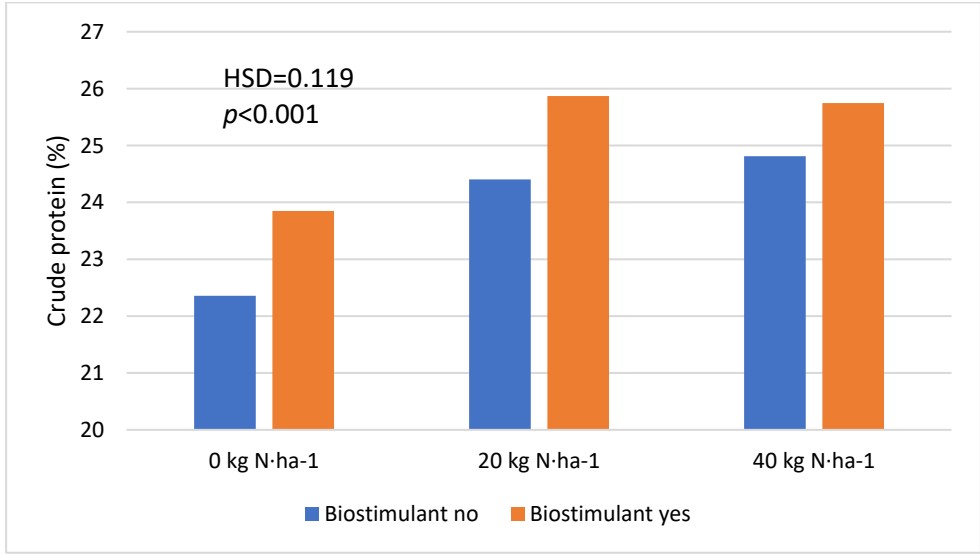

**Figure 3.** Crude protein content in pea seeds after application of nitrogen fertilization and biostimulant. HSD, honestly significant differences.

Application of the biostimulant regardless of the nitrogen rate increased seed protein content, with the higher the nitrogen rate, the smaller the effect of the biostimulant. In the absence of nitrogen fertilization, the increase was 1.5%, but at a dose of 40 kg N·ha$^{-1}$ it was less than 1%. The next figure (Figure 4) illustrates the interaction of biostimulants and nitrogen fertilization factors shaping the content of crude fat in pea seeds (the component that most influences the energy value of seeds). In general, seeds from sites where no biostimulant was applied but different fertilizer nitrogen rates did not differ in crude fat content. The introduction of a biostimulant clearly increases the content of this component in seeds, but the increase is greater in the presence of nitrogen fertilization. The application of nitrogen fertilization at a dose of 40 kg N·ha$^{-1}$ and a biostimulant seems to be superfluous to the fertilization of 20 kg N·ha$^{-1}$ plus a biostimulant, as the fat content of the seeds did not statistically change. The interaction of irrigation and nitrogen fertilization on the level of crude fiber in pea seeds was also found to be significant (Figure 5).

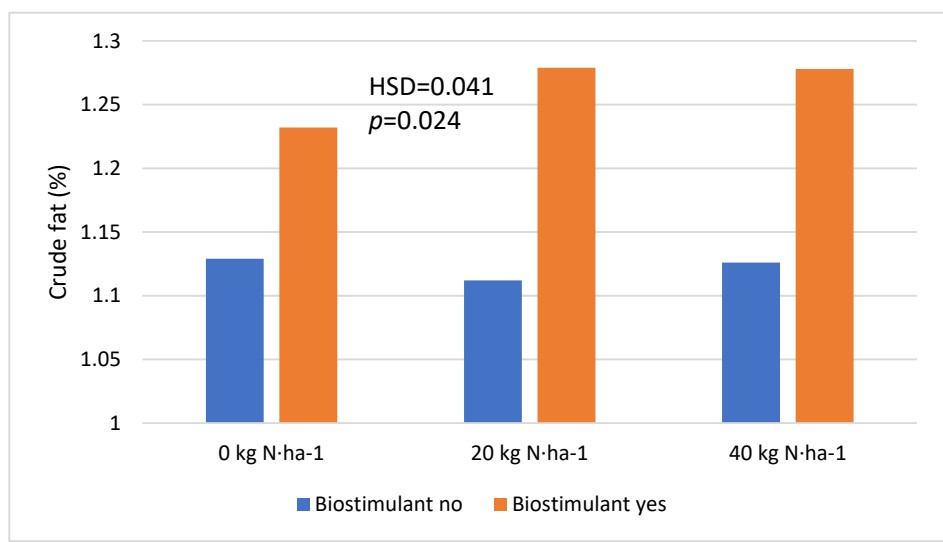

**Figure 4.** Crude fat content in pea seeds after application of nitrogen fertilization and biostimulant. HSD, honestly significant difference.

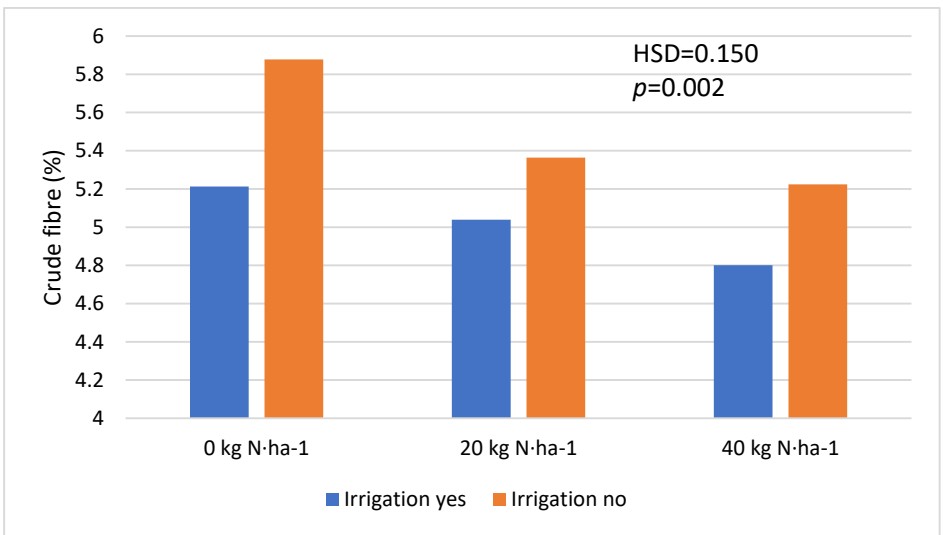

**Figure 5.** Crude fiber content in pea seeds after application of nitrogen fertilization and irrigation. HSD, honestly significant difference.

In the absence of nitrogen fertilization, a higher content of this component can be clearly seen on non-irrigated objects. Of course, such a relationship also persists on nitrogen-fertilized objects, but the decrease in fiber content after the introduction of irrigation is smaller. It should be noted, however, that we relate the decreases to smaller baseline values (nitrogen-fertilized and non-irrigated objects).

Breeders and producers of slaughtered animals increasingly often reach for legume sources of protein, such as pea seeds. McNeill et al. [86] evaluated the effectiveness of using pea seeds at 10% and 20% in broiler chickens' feed. Feed consumption by animals in the control group and those receiving a mixture with 10% pea seeds was the same. An increase in the proportion of pea seeds to 20% in the feed resulted in a significant decrease in feed intake. Janocha et al. [87] suggested that partial replacement of soybean meal with pea meal may be beneficial in the nutrition of broiler chickens due to the fact that it improves rearing performance, carcass composition, and muscle fatty acid profile. Authors recommended to improve the growth performance of broiler chickens to use 10/20% pea meal as a substitute for soybean extraction meal in starter/grower rations, respectively. However, when taking into account the fat quality of the leg muscles, it is recommended to use 10/20% or 15/25%

pea meal in starter/grower diets, respectively. Diaz et al. [88], on the other hand, used a much higher proportion of peas in chicken feed (30%) and reported no negative effect on production performance. Pea seeds can also serve as a protein component for laying hens. The introduction of up to 20% pea seeds in the feed, supplemented with an 8% addition of rapeseed meal, can replace SBM [89]. Halle [90] considers it reasonable to use both of these components in feed for laying hens. According to Ciurescu and Pana [91], there is no need to supplement enzyme preparations in the ration, as the authors noted no significant differences in production performance or egg quality (with 35% pea seed meal in the feed) between the control and experimental groups. Substitution of SBM with pea seeds had no effect on egg quality. Koivunen et al. [30] believe that the introduction of even 30% of pea seeds into the feed does not have a negative impact on the quality of hens and eggs. The use of pea seeds to produce protein concentrates for poultry can also be an alternative to organic production in the face of the current ban on synthetic amino acids [92]. Pea seeds can also be used in the production of compound feed for broiler guinea fowl. Laudadio et al. [22], used pea seeds as a substitute for SBM, obtained similar bird production results as in the control group, while achieving improved carcass quality and a favorable lipid profile. Peas can also be introduced into feed mixes for slaughtered turkeys [93].

## 4. Conclusions

The pea is usually grown in temperate regions, but is accepted as a source of nutrients around the world. The assessed cv. Cysterski was characterized by an average protein content of 25% over a two-year period. The presented own research proved that the protein content in pea seeds increases after application of the biostimulant, both after irrigation and without. Also, the use of a biostimulant, regardless of the amount of nitrogen dose, increased the protein content in seeds, and the higher the dose of nitrogen, the lower the effect of the biostimulant. The energy value of the feed is mainly determined by the fat content. Seeds from objects on which the biostimulant was applied were characterized by a significantly higher content of this component. In addition, it was shown that the application of nitrogen fertilization at a dose of 40 kg·ha$^{-1}$ and a biostimulator seems to be unnecessary compared to fertilization of 20 kg N·ha$^{-1}$ plus a biostimulant, because the fat content in the seeds did not change statistically. Crude fiber is an important component of plant-derived feed, affecting its nutritional value, productivity, and animal health. The use of nitrogen fertilization caused a decrease in the content of crude fiber in relation to that found in seeds from the control object. However, the use of irrigation resulted in a significant increase in fiber concentration. The evaluated pea cultivar contained, on average, 5.3% of crude fiber in seeds. Statistically significant differences in the crude fiber content in pea seeds were found under the influence of the interaction of irrigation and nitrogen fertilization. In the absence of nitrogen fertilization, a higher content of this component is clearly visible on non-irrigated objects. The energy value of the tested pea variety for poultry, estimated based on equations of AME$_n$, was on average 11.94 MJ·(1000 g)$^{-1}$ of seeds. The use of the biostimulant significantly increased the energy value of pea seeds. The results of the experiment allowed us to broaden our knowledge about pea seeds. In a further experiment, seed chemical analyzes will be expanded to include other nutrients.

**Supplementary Materials:** The following supporting information can be downloaded at: https://www.mdpi.com/article/10.3390/agriculture13020376/s1, Table S1. Scheme of field experiment in split-split-plot design.

**Author Contributions:** Conceptualization, C.P., W.B. and R.W.; methodology, W.B., C.P. and R.W.; software, R.W. and S.S.; validation, W.B., C.P., R.W., J.K.-P. and S.S.; formal analysis, R.W., S.S. and W.B.; investigation, C.P. and W.B.; resources, W.B. and C.P.; data curation, W.B., C.P. and R.W.; writing—original draft preparation, W.B., C.P., R.W. and J.K-P.; writing—review and editing, W.B., C.P., R.W. and J.K.-P.; visualization, W.B., R.W. and J.K.-P.; supervision, W.B. and C.P.; project administration, W.B., C.P. and R.W.; funding acquisition, W.B. and C.P. All authors have read and agreed to the published version of the manuscript.

**Funding:** This research received no external funding.

**Institutional Review Board Statement:** Not applicable.

**Data Availability Statement:** Not applicable.

**Conflicts of Interest:** The authors declare no conflict of interest.

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
