# Peer review of "Effect of Irrigation, Nitrogen Fertilization and Amino Acid Biostimulant on Proximate Composition and Energy Value of Pisum sativum L. Seeds"

_agriculture, doi:10.3390/agriculture13020376_

Round 1
Reviewer 1 Report
1-The introduction section is too long. There are so many details that aren’t needed. I have a hard time understanding why this research is conducted and what the problem is being solved. The objectives need to be stated, and the hypothesis needs to be better justified to reveal the importance of this research.
2-To assess the suitability of the pea seeds as poultry feedstuff, you need to conduct an in vivo experiment with poultry in which you feed peas at increasing levels and evaluate live performance and processing performance, determine amino acid digestibility and AME… etc. and eventually estimate the maximum inclusion level of pea in the diet. It isn’t enough to just estimate the AME from an equation that may or may not be accurate and conclude this cultivar of peas can be fed to poultry.
3-I don’t recommend using a general statement in the title (i.g., selected biological and agricultural conditions), and the title should be rewritten to exactly reflect the objectives of this study which evaluated the effects of 3 factors (irrigation, nitrogen fertilization and amino acid biostimulator) or you could write the title based on some important results.
4-In the Statistical analysis and results, it is confusing to report the least significant differences (LSD), although the means were separated using the Tukey test. The LSD test is known as Fisher’s LSD, completely different from the Tukey test. Tukey uses the Minimum Significant Difference, not the least significant of differences. Also, the statistical model should be stated in the Statistical analysis section.
5-Table 1: The LSD (α) is confusing here. The term LSD should be replaced with Minimum Significant Difference to differentiate it from Fisher’s test. Also, alpha is different from the p-value you get from a statistical test. Alpha is the significance level which is 0.05. What you mean is the p-value. The p-value cannot be < 0.000! you should report it as <0.001 not < 0.000. Also, you only presented the main effect means of the 3 factors. The simple effect means (2X3X2 = 12 trt combinations) were not presented!
Table1: the unit for the AME should be reported. The values appeared to be kcal per 100 g, and I suggest converting them to kcal per 1000 g for ease of comparison with stated values elsewhere. Usually, ME values are stated as kcal/kg or MJ/kg. Also, these values are “estimated” values based on equations, not “determined” values based on in vivo assays, and this should be indicated.
6-The p-values for the two- and three-way interactions should be presented. Looking at Figures 2-5, it is not clear to know if there is a significant interaction or not without seeing the p-value.
7-Sometimes the Discussion section seems like a literature review rather than a discussion of the results presented. For example, lines 335-364 have nothing to do with the results being presented. The authors evaluated specific objectives that don’t include feeding peas to chickens and then evaluating the performance or digestibility or amino acid contents of peas. .
8-The conclusions section should be rewritten to reflect the most important findings of this research. There are many claims that are not supported by the results. For example, when saying, “Pea seeds can therefore be successfully used in poultry nutrition,” this implies that you fed the peas to poultry and evaluated the performance and the maximum safe level, which wasn’t done in this research.
Others:
Replace “metabolic energy” with “metabolizable energy”. There is nothing called “metabolic energy”.
Author Response
Response to Reviewer Comments
Dear Reviewer,
We would like to kindly thank you for the insightful review of our manuscript. Below we attached the list of changes made according to your suggestions. In the revised version of the manuscript we have marked the corrected parts of the text in the track change mode.
In behalf of co-authors, once again thank you for your valuable comments.
Wioletta Biel
Reviewer:
Comments to the Author
Point 1: The introduction section is too long. There are so many details that aren’t needed. I have a hard time understanding why this research is conducted and what the problem is being solved. The objectives need to be stated, and the hypothesis needs to be better justified to reveal the importance of this research.
Response: As suggested, we made changes and, above all, shortened the introduction. An economic and research problem is the disturbed protein balance, e.g. in Europe because of the reluctance of the food and feed economy to soybean meal (SBM) from a genetically modified (GM) soybean. Documenting the possibility of replacing (at least partially) the above-mentioned soybean meal with pea meal in animal nutrition, as a result of documenting its similar nutritional value, will be important information for the industry, which may result in a further increase in the cultivation of high-protein crops, especially peas. An equally important element of the research is the assessment of the impact of irrigation, mineral nitrogen fertilization and the use of an amino acid biostimulator on the yield of pea seeds and their chemical composition.
Point 2: To assess the suitability of the pea seeds as poultry feedstuff, you need to conduct an in vivo experiment with poultry in which you feed peas at increasing levels and evaluate live performance and processing performance, determine amino acid digestibility and AME… etc. and eventually estimate the maximum inclusion level of pea in the diet. It isn’t enough to just estimate the AME from an equation that may or may not be accurate and conclude this cultivar of peas can be fed to poultry.
Response: Yes, of course, the authors agree with this, but neither in the title nor in the purpose of the work did we emphasize that we assess the usefulness of pea seeds in poultry nutrition, our goal was, among other things, to indicate whether the use of the tested agricultural conditions has an impact on the chemical composition and AMEn, in the hypothesis, we assumed that the factor having a significant impact on these features is the amino acid biostimulant. And since the estimated energy was energy for poultry, we mention this group of livestock. We really fully agree with what the Reviewer wrote. By emphasizing the value of pea seeds studied in this manuscript, we want it to be even more promoted, to encourage farmers to grow this legume more, as a high-protein feed that can be an alternative (at least partially) to soybean meal.
Point 3: I don’t recommend using a general statement in the title (i.g., selected biological and agricultural conditions), and the title should be rewritten to exactly reflect the objectives of this study which evaluated the effects of 3 factors (irrigation, nitrogen fertilization and amino acid biostimulator) or you could write the title based on some important results.
Response: The title was corrected to „Effect of irrigation, nitrogen fertilization and amino acid biostimulant on proximate composition and energy value of Pisum sativum L. seeds” in accordance with the above suggestion and therefore the keywords were verified so that there were no unnecessary repetitions.
Point 4: In the Statistical analysis and results, it is confusing to report the least significant differences (LSD), although the means were separated using the Tukey test. The LSD test is known as Fisher’s LSD, completely different from the Tukey test. Tukey uses the Minimum Significant Difference, not the least significant of differences. Also, the statistical model should be stated in the Statistical analysis section.
Response: As suggested, the notation LSD, rightly perceived as a difference based on the Fisher's post-hoc test, was replaced with the notation HSD, which is uniquely assigned to the Tukey test (Honestly Significant Difference). Therefore, the expected and necessary changes have been made to the text and tables. In the chapter Statistical analyses, the model of analysis of variance is now presented.
Point 5: Table 1: The LSD (α) is confusing here. The term LSD should be replaced with Minimum Significant Difference to differentiate it from Fisher’s test. Also, alpha is different from the p-value you get from a statistical test. Alpha is the significance level which is 0.05. What you mean is the p-value. The p-value cannot be < 0.000! you should report it as <0.001 not < 0.000. Also, you only presented the main effect means of the 3 factors. The simple effect means (2X3X2 = 12 trt combinations) were not presented!
Response: As noted in the reply to Point 4, the LSD was removed from the entire manuscript. Of course, the value of the probability of rejecting the null hypothesis despite being true (a) was assumed in our research at the level of 0.05. The table shows the test probabilities (p) indicating the size of the deviation from a. As expected, the “a” designation has been changed to “p”. The last suggestion is also true, but adding 1x2, 1x3, 2x3 and 1x2x3 factor interactions to the table would significantly increase it, and it should be noted that most of them were not statistically significant. This action was fully conscious and intentional. Therefore, only significant, research-relevant two-factors interactions were presented in a graphical form (Figures 2-5).
Point 6: Table 1: the unit for the AME should be reported. The values appeared to be kcal per 100 g, and I suggest converting them to kcal per 1000 g for ease of comparison with stated values elsewhere. Usually, ME values are stated as kcal/kg or MJ/kg. Also, these values are “estimated” values based on equations, not “determined” values based on in vivo assays, and this should be indicated.
Response: Thank you for noticing, the unit was added, besides, as suggested, the energy values were converted to kcal per 1000 g, in the work where it was or could suggest that the given energy is based on an experiment on animals, it was corrected to 'estimated', additionally a change was made in abstract (line 26).
Point 7: The p-values for the two- and three-way interactions should be presented. Looking at Figures 2-5, it is not clear to know if there is a significant interaction or not without seeing the p-value.
Response: As expected by the Reviewer, the figures with the p values are now presented.
Point 8: Sometimes the Discussion section seems like a literature review rather than a discussion of the results presented. For example, lines 335-364 have nothing to do with the results being presented. The authors evaluated specific objectives that don’t include feeding peas to chickens and then evaluating the performance or digestibility or amino acid contents of peas.
Response: Thank you for this suggestion, the whole manuscript was corrected a bit, but the authors decided to leave some parts as part of the discussion, besides, other Reviewers did not suggest deletion, but of course, if the Reviewer insists, the authors will remove part of the discussion in the next round.
Point 9: The conclusions section should be rewritten to reflect the most important findings of this research. There are many claims that are not supported by the results. For example, when saying, “Pea seeds can therefore be successfully used in poultry nutrition,” this implies that you fed the peas to poultry and evaluated the performance and the maximum safe level, which wasn’t done in this research.
Response: We reworded the conclusion, and removed the suggested sentences.
Point 10: Replace “metabolic energy” with “metabolizable energy”. There is nothing called “metabolic energy”.
Response: Thank you for this valuable suggestion, we made appropriate changes.

Reviewer 2 Report
lines 36-37. This sentence is awkward. Needs reformulation.
lines 38-39: This sentence is awkward. I do not understand what it going to say. What worsening? We do not use the „post extraction soybean meal” term, it is solvent extracted soybean meal or soybean meal.
lines 48-49 better say organic animal production
lines 52-53: This is not true. There is prohibition to plant GM plants (including soybean), but we are importing GM soybeans as solvent extracted.
line 56: Delete this sentence. This is not true. The problem with alternatives of soybean meal is that that due to the lower yield and protein value, with a comparable production costs, they are usually not competitive with soybean meal. In addition, we never could produce enough protein sources to our animals, because not enough arable lands over the production of cereals, oil seeds, green fodder, etc. Therefore, the introduction so far is in wrong line, needs to be rewritten.
line 62: what high yields mean? What about in comparison to soybean meal on the basis of EUR/ g SID Lys for instance?
line 64-65: This is simply not true. Use feed tables as reference, not papers dealing with other subjects. Simply, peas contain half a protein than soybean meal, impossible to have similar amount of amino acids. I have double checked at feedtables.com
line 66: It is true partially: only if the basis of substitution is based on digestible amino acid content, and still some cases not the whole amount of soybean can be replaced (lines 68-69 confirms that).
lines 68-69: what is whole-feed mixtures?
line 69: better to say proportion instead of amount. In the next sentences some data about the suggested incorporation ratio should be given
line 83: conditioning is preparing to something. It may be better reads: Pea yields highly dependent on ….
line 105: proposal? check grammar
lines 171 and 173: correctly AMEn (n is not capital letter; this is how it is used)
line 175: what is ETEV?
line 178: a graphical description of the split-split-plot design would be informative.
line 180: LSD is the abbreviation of Least Significant Difference, which is a test of group differences (like Tukey). Rephrase the sentence.
line 184: awkward sentence by meaning.
Table 1: The unit of yield is certainly not mg/ha (milligram per hectare), I guess it is ton/hectare. Alpha sign is P (probability), correct. Provide RMSE of models (root mean square error). Restructure to traits by treatments table, and give model P value as well. The result of the range test in a form of superscript letters should be inserted to see that which means are different.
lines 199-200: this is not a clear statistical trend. it is just a trend.
lines 209-210: not completely true
lines 238-239: not true
lines 241-242: not true
line 261: awkward sentence
lines 266-267: this is not true, false information. Small intestinal content is always highly liquid, otherwise enzymes cannot work. Delete.
lines 273-276: this is not true, none of the treatments had significant effect on crude fibre content. The difference is not important from practical considerations.
!!! Every statement on statistical differences needs to be checked, too many errors. !!!
Figures: LSD – not correct, the presented value is most probably the P value of the treatment effect. In that case two value is needed, because there is two factor presented. These figures are repetitions of already presented data in different form, and the discussion is also contains repetition.
The statistical analyses should have tested the interactions as well. Results are missing.
Conclusions are too lengthy, and because of the many errors in results section, I am not convinced about its correctness.
Author Response
Response to Reviewer Comments
Dear Reviewer,
We would like to kindly thank you for the insightful review of our manuscript. Below we attached the list of changes made according to your suggestions. In the revised version of the manuscript we have marked the corrected parts of the text in the track change mode.
In behalf of co-authors, once again thank you for your valuable comments.
Wioletta Biel
Reviewer:
Comments to the Author
Point 1: lines 36-37. This sentence is awkward. Needs reformulation.
Response: Thank you, we rephrased this sentence (lines 39-41).
Point 2: lines 38-39: This sentence is awkward. I do not understand what it going to say. What worsening? We do not use the „post extraction soybean meal” term, it is solvent extracted soybean meal or soybean meal.
Response: We corrected sentence to: “The deepening negative balance of high-protein feed materials is covered by soybean meal (SBM). It should also be noted that the demand for protein in farm animal nutrition has increased. Domestic production incl. legumes, including peas, lupines (white, yellow, blue), field beans and soybeans, cover about 30% of the protein requirement. The remaining 70% comes from imported SBM [1].” (lines 42-46).
Point 3: lines 48-49 better say organic animal production
Response: Thank you, we changed to “organic animal production” (line 56).
Point 4: lines 52-53: This is not true. There is prohibition to plant GM plants (including soybean), but we are importing GM soybeans as solvent extracted.
Response: We corrected to: „ The demand for feed protein in Europe is mainly covered by genetically-modified soybean meal (SBM GM). Due to the growing reluctance of consumers to genetically modified organisms [8] in order to reduce the content of SBM GM in feed, in recent years there has been an intensive search for alternative sources of vegetable protein with a similar content of nutrients. It seems that legume seeds have the greatest potential [5]. The available literature shows that legume seeds contain a relatively large amount of protein [9-10]. For environmental and nutritional reasons, increasing the cultivation area of legumes becomes important [11]. ” (lines 62-69).
Point 5: line 56: Delete this sentence. This is not true. The problem with alternatives of soybean meal is that that due to the lower yield and protein value, with a comparable production costs, they are usually not competitive with soybean meal. In addition, we never could produce enough protein sources to our animals, because not enough arable lands over the production of cereals, oil seeds, green fodder, etc. Therefore, the introduction so far is in wrong line, needs to be rewritten.
Response: Thank you very much for this comment. We deleted suggested incorrectly used sentences and generally taking into account this and other comments of the Reviewer, the introduction was reworded with additional references in many places. We hope that now it will consist of a logical whole.
Point 6: line 62: what high yields mean? What about in comparison to soybean meal on the basis of EUR/ g SID Lys for instance?
Response: We added sentence about SID Lys: “Standardized Ideal Digestibility (SID) values are an important indicator that can help nutritionists formulate broiler diets that better meet the birds' requirements and minimize excess nutrients. For lysine, the SID is similar for both pea and soybean meal, 92% and 91%, respectively [22, 23]” (lines 88-92).
Point 7: line 64-65: This is simply not true. Use feed tables as reference, not papers dealing with other subjects. Simply, peas contain half a protein than soybean meal, impossible to have similar amount of amino acids. I have double checked at feedtables.com
Response: We corrected this part (line 80-86). Of course, the authors confirm that peas contain an average of 25 g of protein in 100 g of DM, and SBM 49-54 g/100 g of DM, respectively. The authors wanted to present the content of amino acids per 100 g of protein, so thank you for this suggestion, because in the current version it could have been unclear to the reader, so we corrected and supplemented the value of amino acids per 100 g of protein based on the data provided in feedtables.com, of course, this confirms the literature data. What's more, the nutritional value of the protein of all species of legume seeds, including peas, is obviously lower than the protein of soybean meal. When evaluating the nutritional value of the protein of these two feeds (per 100 g of protein), it can be seen that the pea seeds amino acids composition is similar to that of soybean meal (feedtables.com) (table below). Compared to soybean meal, pea seeds contain similar amounts of lysine (6.23 and 7.29 g/100 g CP, respectively) and methionine (1.43 and 1 g/100 g CP), which are necessary in poultry or pigs diets. Combining peas with methionine improves the nutritional value of protein and brings it closer to the nutritional value of soybean meal. Among legumes, pea seeds are a better source of protein than lupine or horse bean seeds in terms of amino acid utilization.
These conversions are summarized in a table (data from feedtables.com):
|
Item |
Data |
Item |
Data |
||
|
Pea seed |
Soybean meal |
||||
|
Crude protein (g/100 g) |
20.3 |
Crude protein (g/100 g) |
46.2 |
||
|
EAA |
(g/100 g) |
(g/100 g of protein) |
EAA |
(g/100 g) |
(g/100 g of protein) |
|
Lys |
1.48 |
7.29 |
Lys |
2.88 |
6.23 |
|
Met |
0.20 |
0.99 |
Met |
0.66 |
1.43 |
|
Cys |
0.28 |
1.38 |
Cys |
0.73 |
1.58 |
|
Met+Cys |
0.48 |
2.36 |
Met+Cys |
1.39 |
3.01 |
|
Thr |
0.78 |
3.84 |
Thr |
1.77 |
3.83 |
|
Ile |
0.85 |
4.19 |
Ile |
2.11 |
4.57 |
|
Trp |
0.18 |
0.89 |
Trp |
0.64 |
1.39 |
|
Val |
0.96 |
4.73 |
Val |
2.23 |
4.83 |
|
Leu |
1.45 |
7.14 |
Leu |
3.53 |
7.64 |
|
His |
0.51 |
2.51 |
His |
1.24 |
2.68 |
|
Arg |
1.73 |
8.52 |
Arg |
3.30 |
7.14 |
|
Phe |
0.96 |
4.73 |
Phe |
2.34 |
5.06 |
|
Tyr |
0.63 |
3.10 |
Tyr |
1.63 |
3.53 |
|
Phe+Tyr |
1.59 |
7.83 |
Phe+Tyr |
3.97 |
8.59 |
|
Sum |
49.31 |
Sum |
49.91 |
||
Point 8: line 66: It is true partially: only if the basis of substitution is based on digestible amino acid content, and still some cases not the whole amount of soybean can be replaced (lines 68-69 confirms that).
Response: Thank you, we deleted this sentence.
Point 9: lines 68-69: what is whole-feed mixtures?
Response: We changed to “complete feed mixtures” (line 95).
Point 10: line 69: better to say proportion instead of amount. In the next sentences some data about the suggested incorporation ratio should be given.
Response: Thank you, we changed to “proportion”. We have also added references to support the information in the manuscript (line 96 and 99).
Point 11: line 83: conditioning is preparing to something. It may be better reads: Pea yields highly dependent on ….
Response: We rephrased this sentence (line 110-111).
Point 12: line 105: proposal? check grammar
Response: Thank you for this comment, we changed to “way” (line 134).
Point 13: lines 171 and 173: correctly AMEn (n is not capital letter; this is how it is used)
Response: Thank you for this suggestion, we made appropriate changes.
Point 14: line 175: what is ETEV?
Response: We added the explanation and corrected sentence (lines 209-212).
Point 15: line 178: a graphical description of the split-split-plot design would be informative.
Response: The split-split-plot experimental set-up was additionally described with a mathematical model in the Statistical Analysis chapter, and we propose to include a graphical image of this experimental set-up that has been prepared as Table S1 in the Supplementary Materials.
Point 16: line 180: LSD is the abbreviation of Least Significant Difference, which is a test of group differences (like Tukey). Rephrase the sentence.
Response: In the Statistical Analysis section, the abbreviation LSD has been replaced with HSD, which is uniquely assigned to the Tukey test (Honestly Significant Difference). Necessary corrections have been made to the text and tables due to this change.
Point 17: line 184: awkward sentence by meaning.
Response: We rephrased this sentence (line 237-240).
Point 18: Table 1: The unit of yield is certainly not mg/ha (milligram per hectare), I guess it is ton/hectare.
Response: It was corrected to mg per hectare.
Point 19: Alpha sign is P (probability), correct. Provide RMSE of models (root mean square error). Restructure to traits by treatments table, and give model P value as well. The result of the range test in a form of superscript letters should be inserted to see that which means are different.
Response: As suggested, “has been replaced with "p". Due to the fact that the other Reviewers did not request any changes to the table, we propose to leave it in its current form. The table extends about the description of the differentiation of the averages by adding letter markers in the superscripts of the values of the compared averages.
Point 20: lines 199-200: this is not a clear statistical trend. it is just a trend.
Response: Corrected as expected (lines 255-259).
Point 21: lines 209-210: not completely true
Response: We corrected to: Nitrogen fertilization at 20 kg N·ha-1 also showed a tendency to increase grain yield (p=0.08) (Table 1) (line 270-271).
Point 22: lines 238-239: not true
Response: We decided to leave this sentence because Table 1 confirm this thesis (line 302-304).
Point 23: lines 241-242: not true
Response: We decided to leave this sentence because Table 1 confirm this thesis (line 305-306).
Point 24: line 261: awkward sentence
Response: We removed this sentence (line 325).
Point 25: lines 266-267: this is not true, false information. Small intestinal content is always highly liquid, otherwise enzymes cannot work. Delete.
Response: We deleted this sentence, as suggested (lines 328-329).
Point 26: lines 273-276: this is not true, none of the treatments had significant effect on crude fibre content. The difference is not important from practical considerations.
Response: We changed this sentence to “All the treatments evaluated had a significant effect on the crude fiber content of pea seeds (0.001<p<0.012)” (lines 340-342). It is confirmed by Table 1.
Point 27: !!! Every statement on statistical differences needs to be checked, too many errors. !!!
Response: All statements regarding statistical diversity have been thoroughly analyzed and inaccuracies have been corrected.
Point 28: Figures: LSD – not correct, the presented value is most probably the P value of the treatment effect. In that case two value is needed, because there is two factor presented. These figures are repetitions of already presented data in different form, and the discussion is also contains repetition.
Response: In accordance with the will of Reviewer, additional “p” values have been included in the figures. In order not to cause confusion due to the use of two HSD values (HSDa/b i HSDb/a), the higher of the two is shown in the figures. The figures presented are images of the interaction of factors and the average values presented in them are not included in the table and therefore do not constitute a repetition. The table presents only the analysis of the variability of the main effects (separately irrigation, nitrogen fertilization and the use of a biostymulant).
Point 29: The statistical analyses should have tested the interactions as well. Results are missing.
Response: The analysis of statistically significant interactions is presented in Figures 2-5. Other interactions, as statistically insignificant, were omitted.
Point 30: Conclusions are too lengthy, and because of the many errors in results section, I am not convinced about its correctness.
Response: Thank you for this suggestion, we shortened the conclusion.

Round 2
Reviewer 1 Report
My comments have been taken care of.
Author Response
Response to Reviewer Comments
Dear Reviewer,
We would like to kindly thank you for the insightful review of our manuscript.
In behalf of co-authors, once again thank you for your valuable comments.
Wioletta Biel
Point 1: My comments have been taken care of.
Response: Thank you for appreciating the work we put into improving the manuscript.

Reviewer 2 Report
line 102: 1.43 and 1 is not really similar to me.
line 103: Ileal not Ideal. It comes from the measurement of digestibility at the end of ileum and correction for basal endogenous loss (SID).
line 293: I do not find the word "nasion" appropriate here. It should be feed or dry matter. I checked the reference, but I can not find the factors used for calculation (except digestibility). This needs to be checked and clarified. Calculate energy in MJ not in Kcal, use SI units.
Table 1: I am still not satisfied with the arrangements of Table 1. But, if it acceptable to the editors, I can live with that.
Table 1: HSD (p) column head: HSD I guess is the minimum difference between two means which is significant. Than this is always a number, can not say n.s. (not significant). I think that the order of numbers not correct. It is easier to change the column head to p (HSD). If the treatment is not significant, than you can omit to put 'a' to each mean.
line 548: express energy content in MJ
line 715 express in MJ
Author Response
Response to Reviewer Comments
Dear Reviewer,
We would like to kindly thank you for the insightful review of our manuscript. Below we attached the list of changes made according to your suggestions. In the revised version of the manuscript we have marked the corrected parts of the text in the track change mode.
In behalf of co-authors, once again thank you for your valuable comments.
Wioletta Biel
Reviewer:
Comments to the Author
Point 1: line 102: 1.43 and 1 is not really similar to me.
Response: Thank you, changed to „close” (line 74).
Point 2: line 103: Ileal not Ideal. It comes from the measurement of digestibility at the end of ileum and correction for basal endogenous loss (SID).
Response: Thank you for this valuable suggestion, changed to „Ileal” (line 77).
Point 3: line 293: I do not find the word "nasion" appropriate here. It should be feed or dry matter. I checked the reference, but I can not find the factors used for calculation (except digestibility). This needs to be checked and clarified. Calculate energy in MJ not in Kcal, use SI units.
Response: We changed to „seeds” (line 187). Since the AMEn calculation scheme is given in reference [53] (Janssen, I.W.M.M.A. European table of energy values for poultry feedstuffs, 3rd ed.; Spelderholt, The Netherlands, 1989), the authors have changed the previous reference to this in the material and methods chapter in the appropriate place. Energy has been converted to MJ from kcal according to 1 MJ = 239 kcal (1 kcal = 0.004184 MJ).
Point 4: Table 1: I am still not satisfied with the arrangements of Table 1. But, if it acceptable to the editors, I can live with that.
Response: The authors decided to leave the results in Table 1 in present form, of course we are open to suggestions for changes.
Point 5: Table 1: HSD (p) column head: HSD I guess is the minimum difference between two means which is significant. Than this is always a number, can not say n.s. (not significant). I think that the order of numbers not correct. It is easier to change the column head to p (HSD). If the treatment is not significant, than you can omit to put 'a' to each mean.
Response:
As suggested by the Reviewer, the abbreviation "n.s." was replaced with numerical values. The order of the last column has also been changed from "HSD, p" to "p, HSD". The authors also agree with the last comment suggesting the omission of letter markings confirming or not differentiating the means, but suggest leaving them because it was the will of another Reviewer.
Point 6: line 548: express energy content in MJ
Response: Changed (line 317).
Point 7: line 715 express in MJ
Response: Changed (line 414).
